# Stool-based Xpert MTB/RIF assay for the diagnosis of pulmonary tuberculosis in children at a teaching and referral hospital in Southwest Ethiopia

Mitiku Dubale[1,2,3], Mulualem Tadesse[1,3] *, Melkamu Berhane[4], Mekidim Mekonnen[3], Gemeda Abebe[1,3]

1 Mycobacteriology Research Center, Jimma University, Jimma, Ethiopia, 2 Department of Biology, Faculty of Natural and Computational Sciences, Gambella University, Gambella, Ethiopia, 3 School of Medical Laboratory Sciences, Faculty Health Sciences, Jimma University, Jimma, Ethiopia, 4 Department of Pediatric and Child Health, Faculty Medical Sciences, Jimma University, Jimma, Ethiopia

☯ These authors contributed equally to this work.
* mulualemt.tadesse@gmail.com, mulualem.tadesse@ju.edu.et

**Data Availability Statement:** The dataset used and analyzed in the current study is uploaded as supporting information file (S1 File).

## Abstract

### Background

Diagnosis of tuberculosis (TB) in children is challenging mainly due to the difficulty of obtaining respiratory specimen and lack of sensitive diagnostic tests. The objective of this study was to evaluate the diagnostic performance of Xpert MTB/RIF (Xpert here after) for the diagnosis of pulmonary TB (PTB) from stool specimen in children.

### Methods

A cross-sectional study was conducted among consecutively recruited children (less than 15 years old) with presumptive PTB at Jimma Medical Center, Ethiopia. One pulmonary specimen (expectorated sputum or gastric aspirate) was collected from each participant and tested for TB by Xpert and Lowenstein-Jensen (LJ) culture. In addition, one stool specimen per child was collected and tested by Xpert after a single step, centrifuge-free stool processing method adapted from KNCV TB Foundation. Diagnostic performance of Xpert was calculated with reference to LJ culture and to a composite reference standards (CRS) comprising of confirmed TB (positive by Xpert and/or culture) and unconfirmed TB (clinical diagnosis with improvement after anti-TB treatment).

### Results

A total of 178 children were enrolled; 152 of whom had complete microbiological results. Overall, TB was diagnosed in 13.2% (20/152) of the children with presumptive TB. Of these, only ten had microbiologically confirmed TB (positive Xpert and/or culture) and the remaining ten were clinically diagnosed with positive response to anti-TB treatment and were classified as unconfirmed TB. Stool Xpert had sensitivity of 100% (95%CI: 66.4–100) and specificity of 99.3% (95%CI: 96.2–100) compared to culture; however, the sensitivity was

**Funding:** This work was supported by Jimma University-Mycobacteriology Research Center, Jimma, Ethiopia. The funders had no role in study design, data acquisition, analysis and interpretation, or the decision to prepare the manuscript and submit for publication.

**Competing interests:** The authors have declared that no competing interests exist.

decreased to 50% (95%CI: 27.2–72.8) when compared to CRS. The Xpert on gastric aspirate had sensitivity of 77.8% (95%CI: 40–97.2) compared to culture and 40% (95%CI: 19.1–64) compared to CRS.

## Conclusions

The sensitivity of Xpert for stool sample is comparable to that for gastric aspirate. Stool sample is a potential alternative to pulmonary specimen in the diagnosis of pulmonary TB in children using Xpert.

## Background

Tuberculosis (TB) is one of the leading causes of death globally [1]. Children account for 11% of the global disease burden with an estimated 1.1 million new childhood TB cases and 205,000 pediatric TB deaths in 2018 [1]. Early detection of TB is critical for timely initiation of treatment but is challenging in children because of non-specific clinical presentations, difficulty of obtaining respiratory specimen and lack of sensitive diagnostic tests [2].

Sputum specimen remains being the most frequently used clinical sample to confirm the diagnosis of TB microbiologically [3]. However, in most cases, children are unable to expectorate sputum specimen and when sputum is available; the yield is expected to be poor because of the paucibacillary nature of childhood TB. As a result, specimen obtained through different procedures such as induced sputum, bronchoalveolar lavage, and gastric aspirate have been studied to improve the sensitivity of microbiological examinations [4]. Though these procedures are well tolerated in adults, they are relatively uncomfortable for children, and the capability to carry out these procedures may be lacking in many TB endemic settings. Thus, there is a need for non-sputum based specimen to diagnose TB in children who are unable to expectorate sputum.

Stool is an alternative specimen for TB diagnosis, because *Mycobacterium tuberculosis* (MTB) can be swallowed with the sputum and detected in stool [5]. In particular, stool is easy to obtain from infants and young children who are unable to produce sputum [6]. The introduction of Xpert has revolutionized the diagnosis of TB and the World Health Organization (WHO) has endorsed Xpert for the diagnosis of TB from various types of specimen such as sputum, lymph node tissue and aspirate, cerebrospinal fluid, gastric lavage and aspirate [7]. Recently, studies evaluating Xpert for detection of pulmonary TB (PTB) from stool have been published with highly variable sensitivities, ranging from 32% to 83.3% [6, 8]. In Ethiopia, there is a paucity of data on the diagnostic performance of Xpert on stool specimen for children who are unable to expectorate sputum. Therefore, this study is aimed to evaluate the diagnostic performance of Xpert for the diagnosis of PTB from stool specimen in children with presumptive TB.

## Materials and methods

### Study design and setting

An institution-based cross-sectional study was carried out among consecutively recruited children (less than 15 years old) with presumptive PTB at Jimma University Medical Center (JUMC), Ethiopia between March and November 2019. The study area is located in Jimma Zone, Oromia Regional State in Southwest part of Ethiopia. Jimma town is located 357km away from Addis Ababa (the capital city of Ethiopia). JUMC, located in Jimma town, is one of

the oldest public hospitals in the country. JUMC is a teaching and referral hospital that provides services for about 15 million people in its catchment area. Each year, JUMC reports serving approximately 15, 000 inpatient hospital stays, 160, 000 outpatient visits, and 11, 000 emergency department visits [9]. Most of the patients getting services at the center are from rural area. According to WHO 2021 report, Ethiopia is among the 30 countries with high TB and TB/HIV burden, with an estimated annual incidence of 151,000 cases and an estimated 19,000 TB-related deaths (excluding HIV associated TB deaths).

## Study participants

Participants were children aged less than 15 years presenting with a chronic cough of >2 weeks, weight loss, loss of appetite, persistent fever without an apparent cause, night sweats, or history of close contact with a TB patient within the preceding 12 months. They were classified as the presumptive PTB patients. Participants were excluded, if they had an already confirmed TB, had been on anti-TB treatment (ATT) for >72 hours before enrolment and clinical symptoms or physical signs suggestive of extrapulmonary TB.

## Study procedures

Children up to 15 years of age were screened for TB and those presumptive TB cases were enrolled into this study. TB screening was done as per standard of care, which mainly relies on self-reporting of symptoms suggestive of TB (cough, fever, night sweats, weight loss and history of close contact with TB patients). Study subjects were diagnosed with confirmed TB based on positive microbiological test results (positive LJ culture and/or positive Xpert on gastric aspirate or sputum) or with unconfirmed TB based on a clinical diagnosis and positive response to ATT. Some of the patients with negative microbiologic results were given empirical treatment based on the evaluation by the treating clinicians and were evaluated for response to treatment (resolution of symptoms, weight gain, and radiographic improvement) after two months. Demographic data and clinical information of the participants were collected through structured questionnaires. Clinical data such as radiologic findings, clinical improvement after ATT, HIV status, nutritional status, and BCG vaccination status were obtained from the participants' medical records.

## Specimen collection and laboratory procedures

One respiratory specimen (a minimum of 4ml) and one stool specimen (a minimum of 3gm) were collected per child. Xpert was performed at JUMC Laboratory from respiratory specimens as part of the routine practice at the medical center, whereas TB culture and stool Xpert were performed at Jimma University Mycobacterium Research Center (JUMRC) for the research purposes.

The respiratory samples were collected as per the standard of care. The treating physician collected single gastric aspirate or expectorated sputum. Gastric aspirate samples were collected early in the morning through nasogastric tube following an overnight fasting and tested by Xpert and LJ culture. Expectorated sputum instead of gastric aspirate was used in the patients who were able to produce sputum.

Respiratory specimens were divided into two parts; the first part was diluted (2:1v/v) with Xpert sample reagent, followed by vigorous shaking and incubation at room temperature for 15 minutes. Then, 2ml of liquefied specimen was aspirated using sterile pipette provided with Xpert kit and added to Xpert cartridge and loaded in to the GeneXpert instrument to run the test. The results were reported to the treating physicians as soon as possible for participants' management.

The second part of respiratory specimen was neutralized with phosphate buffered solution (PBS) immediately after collection and stored at 4–8 ˚C for 2–3 days until processing for culture. Equal volume of N-acetyl-L-Cysteine and 4% sodium hydroxide solution were added to the 50ml specimen tube. After mixing on the vortex, the sample tube was left standing for 20 minutes at room temperature for liquefaction and decontamination. Then, sterile PBS (pH6.8) was added up to 45ml to neutralize the alkaline solution, followed by centrifugation at 3000g for 15 minutes. After discarding the supernatant, the pellet was re-suspended in 2ml PBS and two drops of the resulting diluted deposit was inoculated on the slants of LJ medium.

Stool specimen (approximately 3-5g) was collected in a wide mouthed specimen collection jar on spot or submitted the following day and was stored at −20˚C until processed as described previously [10]. Stool Xpert was performed by single step, centrifuge-free stool processing protocol adapted from KNCV TB foundation [10]. Briefly, 1 g of thawed stool was transferred to 50ml falcon tube using wooden applicator stick; 8ml of sample reagent was poured in the sample tube and mixed very well, and left undisturbed for 20 minutes at room temperature. The supernatant was carefully aspirated and added to Xpert cartridge using sterile pipette.

### Diagnostic classification for analysis

Children were classified into three categories based on their clinical, radiological, and laboratory results. (i) "Confirmed TB":–child has symptoms suggestive of TB and TB disease is confirmed microbiologically (positive gastric aspirate Xpert and/or culture); (ii) "Unconfirmed TB":- child has at least 2 of the following: symptoms suggestive of TB, chest radiograph consistent with TB, or documented exposure to MTB and positive response to ATT but TB disease is not confirmed microbiologically (negative gastric aspirate Xpert and/or culture); (iii) "Unlikely TB":- no criteria for "Unconfirmed TB" was met and TB is not confirmed microbiologically (negative gastric aspirate Xpert and/or culture). CRS is defined as either confirmed TB or unconfirmed TB in the definition of TB and cases that met "unlikely TB" criteria were classified as not TB.

### Data analysis

Data were entered in Epidata version 3.3 and analyzed using SPSS software package version 20. The Xpert sensitivity, specificity, positive, and negative predictive values and their 95% confidence intervals (95%CI) were calculated using the following as one of the reference standards: i) LJ culture alone; ii) LJ culture and/or Xpert (confirmed TB); and iii) composite reference standard (CRS) (confirmed TB and Unconfirmed TB). Differences and similarities between the two methods (stool Xpert versus gastric aspirate Xpert) was determined based on the 95%CI. Non-over lapping 95% CIs indicated a difference between the two methods (stool Xpert versus gastric aspirate Xpert) and vice versa.

### Ethical consideration

Ethical approval (Protocol number IHRPGD552/18) was obtained from Institutional Review Board of Jimma University, Ethiopia. We obtained written informed consent from parents or legal guardians of children and assent from children above ten years old.

## Results

### Characteristics of study participants

A total of 178 children with presumptive PTB were enrolled to the study. Out of these, 26 children were excluded from the final analysis: 17 were unable to provide stool specimen and 9

**Table 1. Demographic characteristics of participants with their diagnostic results.**

| Variable | Culture positive n (%) | Culture negative n (%) | GA Xpert positive n (%) | GA Xpert negative n (%) | Stool Xpert positive n (%) | Stool Xpert negative n (%) | P-value |
|---|---|---|---|---|---|---|---|
| **Age (years)** | | | | | | | |
| < 1 | 1(0.7) | 13(7.2) | 1(0.7) | 11(7.2) | 1(0.7) | 11(7.2) | 0.7 |
| **1–4** | 6(3.9) | 73(48) | 5(3.3) | 74(48.7) | 7(4.6) | 68(44.7) | |
| **5–10** | 1(0.7) | 38(25) | 1(0.7) | 38(25) | 1(0.7) | 36(23.7) | |
| **11–14** | 1(0.7) | 21(13.8) | 1(0.7) | 21(13.8) | 1(0.7) | 20(13.2) | |
| **Gender** | | | | | | | |
| **Male** | 0 | 74(48.7) | 1(0.7) | 73(48) | 1(0.7) | 68(44.7) | 0.03 |
| **Female** | 9(5.9) | 69(45.4) | 7(4.6) | 71(46.7) | 9(5.9) | 67(44.1) | |
| **Residence** | | | | | | | |
| **Urban** | 2(1.3) | 52(34.2) | 1(0.7) | 53(34.9) | 2(1.3) | 49(32.2) | 0.4 |
| **Rural** | 7(4.6) | 91(59.9) | 7(4.6) | 91(59.9) | 8(5.30) | 86(56.6) | |

GA = gastric aspirate

had contaminated culture results. We included the remaining 152 participants for whom we analyzed 17 expectorated sputum specimens, 135 gastric aspirates and 152 stool specimens. Seventy-eight (51.3%) of 152 participants were females and participants' ages ranged from 7 months to 14 years (median 3 years). Majority, 98 (64.5%), of the participants were rural residents (Table 1). Regarding the clinical manifestation of the participants, 141 (92.8%) had cough for >2 weeks, 122 (80.3%) had loss of appetite, 109 (71.7%) had fever, 90 (59.2%) had weight loss, 90 (55.9%) had weakness/fatigue, 70 (46.1%) had shortness of breath and 35 (23%) had TB contact history. The majority, 142 (93.4%), were vaccinated with BCG. Close to a third of the participants, 54 (35.5%), were severely malnourished, whereas 6 (3.9%) of them were on anti-retroviral therapy.

## Diagnosis of TB

Out of the 152 pulmonary specimens analyzed, MTB was detected in 9 (6.7%) by culture and in 8 (5.9%) by Xpert. In one of the participants, gastric aspirate Xpert was positive whereas the culture from the same specimen was negative. On the other hand, in two of the participants, culture from the gastric aspirates were positive whereas Xpert results were negative. All the 17 expectorated sputum specimens were negative by both Xpert and culture. Stool Xpert testing revealed 10 (6.6%) MTB positive cases. Rifampicin resistance was not detected in any of the Xpert positive specimens. Table 2 shows results of gastric aspirate Xpert, stool Xpert and gastric aspirate culture amongst study subjects.

Overall, 10 (6.6%) of the 152 study participants had microbiologically confirmed PTB, defined as a positive result on culture and/or Xpert on pulmonary specimen. We reviewed the medical records of the remaining 142 (93.4%) of participants who were microbiologically negative. Of these, 8.5% (12/142) were clinically diagnosed as TB by the clinicians. Among the 12 clinically diagnosed TB cases, 83.3% (10/12) showed clinical improvement after ATT. The remaining 2 cases died after two weeks of ATT initiation and were classified as unlikely TB cases. Out of the 10 cases who have improved clinically, 4 had radiological findings suggestive of TB whereas 6 had no radiological evidences of TB documented on their medical records and hence were classified as "unconfirmed TB". However, none of the children with clinically-diagnosed TB (unconfirmed TB) had positive stool Xpert result. In the remaining 91.5% (130/

**Table 2. Proportion of positive GA Xpert and stool Xpert compared to GA culture for MTB.**

| GA Xpert | GA Culture | | | p-value |
|---|---|---|---|---|
| | Positive n(%) | Negative n(%) | Total n(%) | |
| Positive | 7(4.6) | 1(0.7) | 8(5.3) | 0.00 |
| Negative | 2(1.3) | 142(93.4) | 144(94.7) | |
| Total | 9(5.9) | 143(94.1) | 152(100) | |
| Stool Xpert | GA Culture | | | p-value |
| | Positive n(%) | Negative n(%) | Total n(%) | |
| Positive | 9(6.2) | 1(0.7) | 10(6.9) | 0.00 |
| Negative | 0 | 135(93.1) | 135(93.1) | |
| Total | 9(6.2) | 136(93.8) | 145(100) | |

MTB = *Mycobacterium tuberculosis*, GA = gastric aspirate

142) of the cases, TB was ruled out and an alternative diagnosis was made and hence, they were classified as "unlikely TB" (Fig 1).

## Diagnostic performance of stool Xpert

Error/invalid results were documented in 4.6% (7/152) of Xpert tests performed on stool. Due to a shortage of Xpert cartridges, we didn't repeat the tests in these invalid cases. Of the 145 stool specimens with valid Xpert results, 6.9% (10/145) were MTB positive. Stool Xpert detected all gastric aspirate culture confirmed TB cases and one positive case missed by gastric aspirate culture (Table 2). Moreover, stool Xpert also detected two MTB positive cases which were found to be negative on gastric aspirate Xpert.

Using culture of the respiratory specimen (gastric aspirate and expectorated samples) as the reference standard, stool Xpert had sensitivity of 100% (95% CI: 66.4–100) and specificity of 99.3% (95% CI: 96.2–100), whereas gastric aspirate Xpert had sensitivity of 77.8% (95% CI: 40–97.2) and specificity of 99.3% (95% CI: 96.2–100) (Table 3). Using gastric aspirate culture and/or Xpert positivity (microbiological confirmation) as one of the reference standards, stool Xpert had each 100% sensitivity, specificity, PPV and NPV (Table 4 and S1 Table).

Moreover, the sensitivity, specificity, PPV, and NPV of stool and gastric aspirate Xpert were also calculated with reference to CRS. Accordingly, stool Xpert had a sensitivity of 50% (95% CI; 27.2–72.8), specificity of 100% (95% CI: 97.1–100), PPV of 100% and NPV of 92.6% (95% CI; 89–95.1) against CRS (Table 5). The corresponding sensitivity, specificity, PPV and NPV for gastric aspirate Xpert were 40% (95% CI; 19.1–64), 100% (95% CI; 97.2–100), 100% and 91.7% (95% CI; 88.5–94) respectively compared to CRS (Table 5). Stool and gastric aspirate Xpert MTB detection rates compared to CRS is also shown in supporting information (S2 Table).

## Discussion

Ethiopia is among the 30 countries with high-burden of TB and TB/HIV in the world and there is a huge number of undiagnosed TB in children [11]. The national guidelines for TB care in Ethiopia recommends Xpert for the diagnosis of childhood TB. However, obtaining appropriate respiratory specimens from children is difficult. Thus, a non-invasive sample for the diagnosis of TB in children would improve care for this population. In the current study, we have demonstrated that stool Xpert has comparable performance with gastric aspirate Xpert in consecutively recruited children with presumptive PTB. Other studies have also reported promising findings [6, 12].

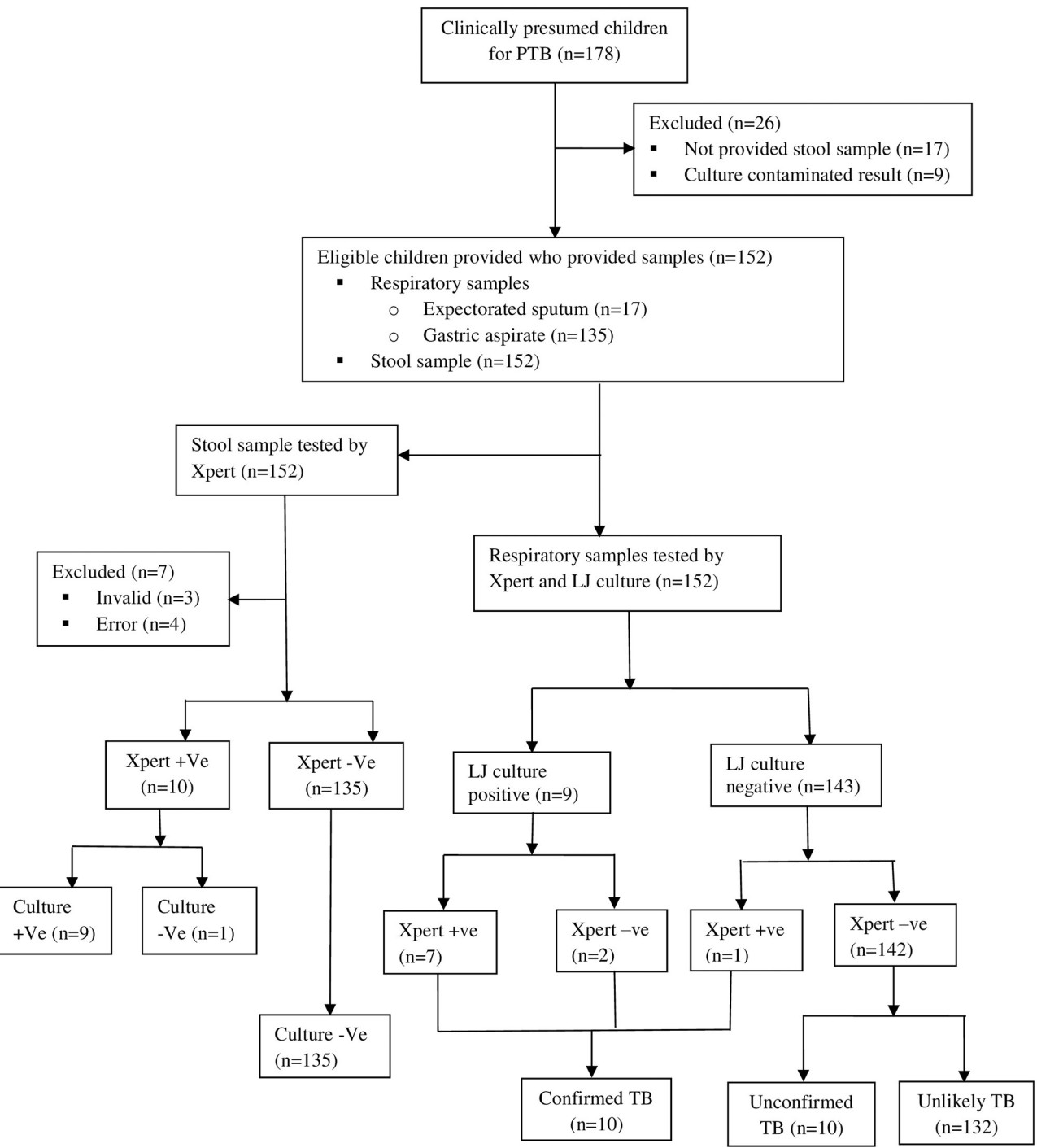

**Fig 1. Flow chart describing study workflow and diagnostic classifications.** TB = tuberculosis, PTB = pulmonary tuberculosis, n = number, +Ve = positive, -Ve = negative, LJ = Lowenstein-Jensen.

We found that Xpert was 100% sensitive for MTB detection in stool among children with microbiological confirmation from a respiratory specimen and 50% sensitive in children with microbiologically and/or clinically diagnosed TB. Other studies [8, 13, 14] conducted in Ethiopia, Kenya and Pakistan have also reported excellent sensitivity (ranging from 88.9% to 100%)

**Table 3. Diagnostic performance of GA Xpert and stool Xpert using culture as a reference standard.**

| Tests | Diagnostic performances | | | |
|---|---|---|---|---|
| | Sensitivity % (95%CI) | Specificity % (95%CI) | PPV% (95%CI) | NPV % (95%CI) |
| GA Xpert | 77.8(40–97.2) | 99.3(96.2–100) | 87.5(49–98.1) | 98.6(95.5–99.6) |
| Stool Xpert | 100(66.4–100) | 99.3(96.2–100) | 90(56.1–98.5) | 100 |

GA = gastric aspirate, CI = confidence interval, PPV = positive predictive value, NPV = negative predictive value

of stool Xpert compared to gastric aspirate culture. The lower sensitivity of stool Xpert compared to CRS (microbiological and/or clinically confirmed TB) is possibly due to the presence of paucibacillary disease in patients clinically diagnosed for TB.

The specificity of stool Xpert in our study when compared to culture was consistent with other previous studies conducted in Ethiopia, South Africa and Egypt which have demonstrated specificities ranging from 99.3 to 99.7% [6, 8, 15]. The specificity of stool Xpert in our study when compared to CRS was similar with another study conducted in Pakistan [14]. One culture negative gastric aspirate specimen was found to be positive on both stool and gastric aspirate Xpert tests. This may be due to the presence of non-viable bacilli during decontamination process of gastric aspirate specimen.

In this study, stool Xpert identified all of the gastric aspirate Xpert positive cases, suggesting stool as a potential alternative for use to gastric aspirate in the routine diagnosis of PTB. Gastric aspiration may be used to retrieve pulmonary specimen from patients who cannot expectorate sputum but it is an invasive procedure and needs trained health workers. As a non-invasive sample, stool is considered to be safe, easy for collection and has the potential to be used for detection of MTB. Moreover, stool Xpert test is performed by simple stool testing protocol adapted from KNCV TB foundation [11], which omits some labor intensive steps such as homogenization, decontamination and centrifugation of stool specimen prior to Xpert testing done by other studies [13–15]. This approach significantly reduces sample processing time, minimizes the workload on the laboratory personnel and also minimizes costs.

In our study, unfortunately, none of the clinically diagnosed TB cases were detected by stool Xpert. Poor performance of stool Xpert in the clinically diagnosed (microbiologically negative) children has also been reported in other studies [15, 16]. This could potentially indicate the limitation of the stool Xpert since patients with such clinically-diagnosed unconfirmed TB are more likely to have paucibacillary disease. Children with a high clinical probability of TB despite a negative stool Xpert should be started on ATT until better and more sensitive tools are available.

Older children are able to expectorate sputum and may produce more adult-type sputum. However, in our study, none of the expectorated sputum samples were positive for *M. tuberculosis*. This could be due to the poor sample quality in the expectorated sputum samples.

**Table 4. The diagnostic performances of stool Xpert and GA Xpert using LJ culture and/or GA Xpert (microbiological confirmations) as one of the reference standards.**

| Tests | Diagnostic performances | | | |
|---|---|---|---|---|
| | Sensitivity % (95%CI) | Specificity % (95%CI) | PPV % (95%CI) | NPV % (95%CI) |
| Stool Xpert | 100 | 100 | 100 | 100 |
| GA Xpert | 80(79.2–80.8) | 100 | 100 | 98.6(98.6–98.7) |

GA = gastric aspirate, CI = confidence interval, PPV = positive predictive value, NPV = negative predictive value

**Table 5. The diagnostic performance of GA Xpert, stool Xpert and GA culture compared to CRS.**

| Tests | Diagnostic performances | | | |
|---|---|---|---|---|
| | Sensitivity % (95%CI) | Specificity % (95%CI) | PPV % (95%CI) | NPV % (95%CI) |
| **GA Xpert** | 40(19.1–64) | 100(97.2–100) | 100 | 91.7(88.5–94) |
| **Stool Xpert** | 50(27.2–72.8) | 100(97.1–100) | 100 | 92.6(89–95.1) |
| **GA culture** | 45(23.1–68.5) | 100(97.2–100) | 100 | 92.3(89–94.7) |

GA = gastric aspirate, CI = confidence interval, PPV = positive predictive value, NPV = negative predictive value, CRS = composite reference standard

However, we did not assess the quality of the sputum sample in the current study which might be taken as a limitation.

Our study has some limitations. We used small sample size due to resource constraints. An additional limitation of our study is the fact that, we collected only single respiratory and stool specimen rather than successive specimens which could have probably increased the yield of the tests done.

## Conclusion

The sensitivity of stool Xpert is comparable to that of gastric aspirate Xpert. Stool is a potential alternative to pulmonary specimen in the diagnosis of PTB in patients who cannot produce sputum. Moreover, stool collection is easier and relatively safe compared to pulmonary specimen and can be easily implemented at lowest level of health care system. However, the diagnostic yield of stool Xpert still requires further validation and optimization using larger sample size.

## Supporting information

**S1 Table. The stool Xpert and GA Xpert MTB detection rate compared to LJ culture and/ or GA Xpert positivity (microbiological confirmation).** LJ = Lowenstein-Jensen, MTB = *Mycobacterium tuberculosis*, GA = gastric aspirate.
(DOCX)

**S2 Table. Stool Xpert, GA Xpert and GA culture MTB detection rate compared to composite reference standard (confirmed and unconfirmed TB).** MTB = *Mycobacterium tuberculosis*, GA = gastric aspirate.
(DOCX)

**S1 File. The original SPSS dataset used and analyzed in the current study.**
(SAV)

## Acknowledgments

We would like to thank the study participants who consented to take part in this study. We are also grateful to the staff of Mycobacteriology Research Center and laboratory personnel at JUMC Laboratory for their assistance and guidance during laboratory work and data collection.

## Author Contributions

**Conceptualization:** Mitiku Dubale, Mulualem Tadesse.

**Data curation:** Mitiku Dubale.

**Formal analysis:** Mitiku Dubale.

**Investigation:** Mulualem Tadesse, Melkamu Berhane.

**Methodology:** Mitiku Dubale, Mulualem Tadesse, Melkamu Berhane, Gemeda Abebe.

**Project administration:** Gemeda Abebe.

**Resources:** Gemeda Abebe.

**Supervision:** Mulualem Tadesse, Melkamu Berhane, Mekidim Mekonnen, Gemeda Abebe.

**Writing – original draft:** Mitiku Dubale.

**Writing – review & editing:** Mulualem Tadesse, Melkamu Berhane, Mekidim Mekonnen, Gemeda Abebe.

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
