## [Decision Letter · Decision Letter 0]

9 Aug 2021

PONE-D-21-15359

To: Stool-based Xpert MTB/RIF assay for the diagnosis of pulmonary tuberculosis in children at a Teaching and Referral Hospital in Southwest, Ethiopia

PLOS ONE

Dear Dr. Tadesse,

Thank you for submitting your manuscript to PLOS ONE. After careful consideration, we feel that it has merit but does not fully meet PLOS ONE’s publication criteria as it currently stands. Therefore, we invite you to submit a revised version of the manuscript that addresses the points raised during the review process.

We look forward to receiving your revised manuscript.

Kind regards,

Silvia Chiang

Academic Editor

PLOS ONE

1. Please ensure that your manuscript meets PLOS ONE's style requirements, including those for file naming. The PLOS ONE style templates can be found at https://journals.plos.org/plosone/s/file?id=wjVg/PLOSOne_formatting_sample_main_body.pdf and https://journals.plos.org/plosone/s/file?id=ba62/PLOSOne_formatting_sample_title_authors_affiliations.pdf.

2. No Funding:

"Thank you for stating the following financial disclosure:

“No”

3. Thank you for stating the following in the Acknowledgments/Funding Section of your manuscript:

“This work was supported by Jimma University, Institute of Health and Jimma University-Mycobacteriology Research Center, Jimma, Ethiopia. The funders had no role in study design, data acquisition, analysis and interpretation, or the decision to prepare the manuscript and submit for publication.”

Please remove any funding-related text

 from the manuscript and let us know how you would like to update your Funding Statement. Currently, your Funding Statement reads as follows:

“No”

a. If there are ethical or legal restrictions on sharing a de-identified data set, please explain them in detail (e.g., data contain potentially sensitive information, data are owned by a third-party organization, etc.) and who has imposed them (e.g., an ethics committee). Please also provide contact information for a data access committee, ethics committee, or other institutional body to which data requests may be sent.

b. If there are no restrictions, please upload the minimal anonymized data set necessary to replicate your study findings as either Supporting Information files or to a stable, public repository and provide us with the relevant URLs, DOIs, or accession numbers. For a list of acceptable repositories, please see http://journals.plos.org/plosone/s/data-availability#loc-recommended-repositories.

5. Please note that in order to use the direct billing option the corresponding author must be affiliated with the chosen institute. Please either amend your manuscript to change the affiliation or corresponding author, or email us at plosone@plos.org with a request to remove this option.

7. Your ethics statement should only appear in the Methods section of your manuscript. If your ethics statement is written in any section besides the Methods, please delete it from any other section.

Additional Editor Comments (if provided):

In addition to addressing the reviewer's comments below, please also address the following issues:

1. I agree with the reviewer's first two major comments and would also add that the authors should explicitly explain how they use this classification in their analysis under the "Data analysis" paragraph.

2. Abstract/Methods, line 32 and all subsequent references to the composite reference standard (CRS): I would be better to say "clinical diagnosis with improvement after anti-TB treatment" instead of just "clinical improvement after anti-TB treatment."

3. Introduction, line 65: What do you mean by "most significant clinical sample?" The "most frequently used?" Please be more specific.

4. Introduction, lines 66-70: The yield is not poor because of many young children's inability to expectorate, but it is poor because of the paucibacillary nature of most childhood TB. Children's inability to expectorate means that other methods to obtain specimens are needed.

5. Introduction, line 71: The collection of sputum is not unsafe for children. These procedures are well tolerated; they are just resource-intensive, as the authors said, and also the capability to carry out these procedures may be lacking in many TB endemic areas. Please either justify or take out this reference to the procedures being unsafe.

6. Methods: Please add a sentence about the incidence of TB in Ethiopia, and please also briefly describe the setting of the study site (e.g. is it urban or rural? are most patients publicly insured? from rural settings? etc.)

7. Methods, lines 94-96: Please specify that these participants were classified as the presumptive TB patients.

8. Methods: Could the authors specify whether a minimum volume was collected for each respiratory sample?

9. Methods, lines 165-167: Do you mean "legal guardians" instead of "caregivers" (just wondering if only parents and legal guardians are allowed to sign informed consent)? Also, what was the minimum age for obtaining assent from the child?

10. Results, line 189: Please replace ". . . not detected in all . . ." with ". . . not detected in any . . ."

11. Results and Discussion: In Methods, the authors state that one of the reference standards is positive culture; I assume this refers to positive culture from any respiratory specimen, not just gastric aspirate. I understand that none of the expectorated sputum samples were positive for M. tuberculosis. However, instead of referencing microbiological confirmation from GA as the reference standard, please reference microbiological confirmation from a respiratory specimen.

12. In addition, I find it odd that none of the expectorated sputum samples were positive for M. tuberculosis. I would expect the older kids to be able to expectorate sputum, and they tend to have more adult-type TB. Could the authors provide an explanation? Were there a lot of epithelial cells indicating poor sample quality in the expectorated sputum samples?

13. Figure 1 is good but could be improved with a few minor changes. I would take away the fist box "children visit JMC." I'd use the same term "presumptive TB cases" in the second box to be consistent w/ the rest of the manuscript. In the "results available" box I would take out the reference to the 10 that improved on ATT and instead add that to the culture negative  Xpert negative part (so you can indicate how many were classified as not TB and how many were classified as unconfirmed TB). I would also take out the box that says "152 patients were included in the analysis" because it's repetitive.

Reviewers' comments:

Reviewer's Responses to Questions

**Comments to the Author**

1. Is the manuscript technically sound, and do the data support the conclusions?

Reviewer #1: Partly

2. Has the statistical analysis been performed appropriately and rigorously? 

Reviewer #1: Yes

3. Have the authors made all data underlying the findings in their manuscript fully available?

Reviewer #1: Yes

4. Is the manuscript presented in an intelligible fashion and written in standard English?

Reviewer #1: Yes

5. Review Comments to the Author

Reviewer #1: The authors evaluated the performance of stool-based Xpert MTB/RIF testing for childhood TB at a referral hospital in Ethiopia using the KNCV stool processing method. The study is of importance in the field given the need for non-sputum, non-invasive approaches to childhood TB, and the manuscript is well-written. The main limitation of the paper is that there were only 10 culture-confirmed TB cases and 10 clinically-diagnosed cases, which prevents clear conclusions on the performance of stool-based testing; however, these initial findings are promising and important to share. Areas to address to improve this manuscript include:

Major

1. The TB classification used was confirmed/probable/possible/not TB; the childhood TB classification was updated in 2015 (Graham et al. CID): https://academic.oup.com/cid/article/61/suppl_3/S179/355883

Adjusting their classification to this structure (Confirmed/Unconfirmed/Unlikely) would be important to allow comparison with other studies and any meta-analyses

2. Similarly, the CRS is defined as a combining Unconfirmed TB with Confirmed TB in the definition of TB, and Unlikely TB as not TB. Again, would adjust definitions accordingly to be consistent with other studies. Would also move that up to the section on diagnostic classification

3. Would note what measures of diagnostic accuracy were used in the methods, including sensitivity, specificity, positive and negative-predictive values

4. In methods, please describe your approach to comparing were sputum and stool-based Xpert. Options include comparing 95% CIs, as well as McNemar’s paired test of proportions. Would be clear in the methods how similarity vs. difference in the two methods are defined, as that is one of your main conclusions.

5. Was smear microscopy performed on the respiratory specimens? If so, please indicate the results and stratify performance smear status

6. Was HIV testing done? Please indicate if done and HIV prevalence

7. In the 2 cases that died, how long were they taking the ATT? Would note the duration, because if soon after initiation, would exclude as unclassifiable

8. In determining the performance of stool Xpert for Confirmed cases, please use the 10 cases (9 culture confirmed, 1 GA Xpert positive) as opposed to 9 culture-positive only

Minor

1. To increase the work’s impact, would note earlier that a single step, centrifuge-free stool processing method was used earlier in the abstract/introduction, as this approach in particular is of interest in the field, including the KNCV kit.

2. There are some spacing issues, such as in line 61 “1.1million” and “in2018” in line 62. Please ensure spacing is okay throughout

3. Were invalid stool results repeated?

4. Please indicate in the results that none of the children with clinically-diagnosed TB (probable/possible) were stool Xpert positive.

6. PLOS authors have the option to publish the peer review history of their article (what does this mean?). If published, this will include your full peer review and any attached files.

Reviewer #1: No

---

## [Author Response · Author response to Decision Letter 0]

4 Nov 2021

Response to Editor and Reviewers

We appreciate the editor and reviewers for the constructive comments which we have used to improve the quality of the manuscript. We have re-written some portions of the manuscript accordingly. We have carefully addressed the comments line by line as follows.

Editor’s comments to the Author:

Response to Editor’s comments:

Comment 1: Please ensure that your manuscript meets PLOS ONE's style requirements, including those for file naming. The PLOS ONE style templates can be found at https://journals.plos.org/plosone/s/file?id=wjVg/PLOSOne_formatting_sample_ main_body.pdf and https://journals.plos.org/plosone/s/file?id=ba62/PLOSOne_formatting_sample_title_authors_affiliations.pdf .

Response 1: We have rechecked our manuscript for the PLOS ONE’s requirement and revised it accordingly. 

Comment 2: Further clarification on source of funding.

Response 2: Thank you for your feedback on the source of funding. No specific funding was obtained for this research work and we have included the following statement in the revised version under the “Funding sources’’ subheading: ‘The authors received no specific funding for this work.’ 

Comment 3: We note that you have provided funding information that is not currently declared in your Funding Statement. However, funding information should not appear in the Acknowledgments section or other areas of your manuscript. We will only publish funding information present in the Funding Statement section of the online submission form. Please remove any funding-related text 

Response 3: No funding information was presented in the acknowledgment part. Regarding comments raised on source of funding, we have addressed them above (Response 2). I feel that we don’t need to update the funding statement. The funding statement can still read as “No”. 

Comment 4: We note that you have indicated that data from this study are available upon request. PLOS only allows data to be available upon request if there are legal or ethical restrictions on sharing data publicly.

Response 4: We have uploaded our data set as a supporting information file (S1 File). We have addressed this in the revised manuscript. 

Comment 5: Please note that in order to use the direct billing option the corresponding author must be affiliated with the chosen institute. Please either amend your manuscript to change the affiliation or corresponding author, or email us at plosone@plos.org with a request to remove this option.

Response 5: As it has been indicated in the manuscript cover page, the corresponding author is affiliated to Mycobacteriology Research Center, Jimma University, Jimma, Ethiopia. 

Comments 6: PLOS requires an ORCID iD for the corresponding author in Editorial Manager on papers submitted after December 6th, 2016. Please ensure that you have an ORCID iD and that it is validated in Editorial Manager. 

Response 6: Here is the ORCID iD of the corresponding author: https://orcid.org/0000-0003-4751-2225 .

Comment 7: Your ethics statement should only appear in the Methods section of your manuscript. If your ethics statement is written in any section besides the Methods, please delete it from any other section.

Response 7: We have checked that ethics statement is mentioned only in the method section under the subheading “Ethical considerations”.

Additional Editor Comments (if provided):

Response to Editor’s comment

Comment 1: I agree with the reviewer's first two major comments and would also add that the authors should explicitly explain how they use this classification in their analysis under the "Data analysis" paragraph.

Response 1: We appreciate the editor and the reviewer for the interesting feedback on TB diagnostic classification. This is well accepted comment and we have adjusted the TB diagnostic classification as it was proposed by Graham et al, 2015 and described it in the methods section under the Diagnostic classification for analysis subheading. 

Comment 2: Abstract/Methods, line 32 and all subsequent references to the composite reference standard (CRS): I would be better to say "clinical diagnosis with improvement after anti-TB treatment" instead of just "clinical improvement after anti-TB treatment."

Response 2: The editor’s feedback was found interesting and we have modified it accordingly. 

Comment 3: Introduction, line 65: What do you mean by "most significant clinical sample?" The "most frequently used?" Please be more specific.

Response 3: We have changed the phrase to “the most frequently used’’.

Comment 4: Introduction, lines 66-70: The yield is not poor because of many young children's inability to expectorate, but it is poor because of the paucibacillary nature of most childhood TB. Children's inability to expectorate means that other methods to obtain specimens are needed.

Response 4: We partly agree with the Editor’s comment. No doubt that sputum is the most frequently utilized sample for the diagnosis of pulmonary TB. However, in most cases, children are unable to expectorate sputum specimen and when sputum is available, the yield is expected to be poor because of the paucibacillary nature of most childhood TB. This is included in the revised manuscript. 

Comment 5: Introduction, line 71: The collection of sputum is not unsafe for children. These procedures are well tolerated; they are just resource-intensive, as the authors said, and also the capability to carry out these procedures may be lacking in many TB endemic areas. Please either justify or take out this reference to the procedures being unsafe.

Response 5: The feedback from the editor is found interesting and we have modified it accordingly.

Comment 6: Methods: Please add a sentence about the incidence of TB in Ethiopia, and please also briefly describe the setting of the study site (e.g. is it urban or rural? are most patients publicly insured? from rural settings? etc.)

Response 6: We have included these information accordingly in the revised version.

Comment 7: Methods, lines 94-96: Please specify that these participants were classified as the presumptive TB patients.

Response 7: We agreed with the Editor’s comment and we have specified accordingly. 

Comment 8: Methods: Could the authors specify whether a minimum volume was collected for each respiratory sample?

Response 8: A minimum volume of 4ml for respiratory specimen and 3gm of stool specimen were collected. We have specified this in the revised manuscript.

Comment 9: Methods, lines 165-167: Do you mean "legal guardians" instead of "caregivers" (just wondering if only parents and legal guardians are allowed to sign informed consent)? Also, what was the minimum age for obtaining assent from the child?

Response 9: We have replaced the term “caregivers” by “legal guardians’’ in the revised manuscript. The minimum age for obtaining assent from children was 11 years age. We have indicated this in the revised manuscript.

Comment 10: Results, line 189: Please replace ". . . not detected in all . . ." with ". . . not detected in any…"

Response 10: We have replaced it accordingly.

Comment 11: Results and Discussion: In Methods, the authors state that one of the reference standards is positive culture; I assume this refers to positive culture from any respiratory specimen, not just gastric aspirate. I understand that none of the expectorated sputum samples were positive for M. tuberculosis. However, instead of referencing microbiological confirmation from GA as the reference standard, please reference microbiological confirmation from a respiratory specimen.

Response 11: The comment is well accepted and we have modified it accordingly. 

Comment 12: In addition, I find it odd that none of the expectorated sputum samples were positive for M. tuberculosis. I would expect the older kids to be able to expectorate sputum, and they tend to have more adult-type TB. Could the authors provide an explanation? Were there a lot of epithelial cells indicating poor sample quality in the expectorated sputum samples?

Response 12: Thank you for the interesting and critical observation. As you rightly stated, older children could expectorate sputum and may produce more adult-type sputum. However, in the current study, none of the expectorated sputum samples was positive for MTB. This could be due to the presence of a lot of epithelial cells indicating poor sample quality in the expectorated sputum samples. However, we didn’t assess the quality of the sputum sample in the current study and we are unable to comment on it. This is addressed in the discussion section. 

Comment 13: Figure 1 is good but could be improved with a few minor changes. I would take away the fist box "children visit JMC." I'd use the same term "presumptive TB cases" in the second box to be consistent w/ the rest of the manuscript. In the "results available" box I would take out the reference to the 10 that improved on ATT and instead add that to the culture

negative  Xpert negative part (so you can indicate how many were classified as not TB and how many were classified as unconfirmed TB). I would also take out the box that says "152 patients were included in the analysis" because it's repetitive.

Response 13: We have modified the figure accordingly. 

Review Comments to the Author

Response to Reviewers’ comments:

Major 

Comment 1: The TB classification used was confirmed/probable/possible/not TB; the childhood TB classification was updated in 2015 (Graham et al.) adjusting their classification to this structure (Confirmed/Unconfirmed/Unlikely) would be important to allow comparison with other studies and any meta-analyses

Response 1: We are very grateful to the editor and the reviewer for the interesting feedback on TB diagnostic classification. As we have mentioned above, this is well accepted comment and we have adjusted the TB diagnostic classification as it was proposed by Graham et al, 2015 and described it in the method section under the Diagnostic classification for analysis subheading.

Comment 2: Similarly, the CRS is defined as a combining Unconfirmed TB with Confirmed TB in the definition of TB, and Unlikely TB as not TB. Again, would adjust definitions accordingly to be consistent with other studies. Would also move that up to the section on diagnostic classification

Response 2: This is also very well accepted comment from the reviewer. In the revised manuscript, this is corrected accordingly.

Comment 3: Would note what measures of diagnostic accuracy were used in the methods, including sensitivity, specificity, positive and negative-predictive values.

Response 3: We agree with the reviewer’s comment and it is indicated as follow in the revised version of the manuscript. “The Xpert sensitivity, specificity, positive, and negative predicted values and their 95% confidence interval (95%CI) were calculated compared to LJ culture and composite reference standard (CRS).’’ We calculated the Xpert sensitivity, specificity, PPV and NPV using LJ culture and CRS as reference standard, though our main conclusion comes from using CRS. 

Comment 4: In methods, please describe your approach to comparing were sputum and stool-based Xpert. Options include comparing 95% CIs, as well as McNamara’s paired test of proportions. Would be clear in the methods how similarities vs. difference in the two methods are defined, as that is one of your main conclusions.

Response 4: This is also a very fascinating feedback from the reviewer. Here we would like to clarify how the diagnostic difference or similarity is determined for the two methods (stool-Xpert vs GA-Xpert) in our study. We have revised our manuscript and the following statements were included “Differences and similarities between the two methods (stool-Xpert versus GA-Xpert) is determined based on the 95%CI. Non-over lapping 95%CI dictated the presence of difference between the two methods and vice versa.’’

Comment 5: Was smear microscopy performed on the respiratory specimens? If so, please indicate the results and stratify performance smear status

Response 5: Smear microscopy was not performed in our study. We expect that the yield from smear microscopy is minimal due to the paucibacillary nature of the disease and we didn’t perform it. 

Comment 6: Was HIV testing done? Please indicate if done and HIV prevalence

Response 6: HIV test was not done. We tried to retrieve HIV test result from patients’ medical records but there were lots of missed results due to poor documentation practice. 

Comment 7: In the 2 cases that died, how long were they taking the ATT? Would note the duration, because if soon after initiation, would exclude as unclassifiable

Response 7: The 2 cases that died were after 2 weeks of ATT initiation. We indicated the duration in the revised version of the manuscript. We thought that if it is TB, they would have shown some clinical improvements to the ATT within 2 weeks of treatment. 

Comment 8: In determining the performance of stool Xpert for Confirmed cases, please use the 10 cases (9 culture confirmed, 1 GA Xpert positive) as opposed to 9 culture-positive only

Response 8: Many studies used culture alone as a reference standard when determining the diagnostic accuracy of new or already available diagnostic tools. Culture is one of the best method to detect TB and many scholars considered culture as the gold standard method. However, for pediatric and extra-pulmonary form of TB, culture alone may not be used as a reference standard as it misses a significant number of TB cases due to the paucibacillary nature of these disease. Our main interest here is to determine the performance of stool-Xpert using culture alone as a reference standard and to compare it with the diagnostic values when CRS is used as a reference standard. However, the main conclusion of our study was derived based on Stool-Xpert performance using CRS as a reference standard. 

Minor

Comment 1: To increase the work’s impact, would note earlier that a single step, centrifuge-free stool processing method was used earlier in the abstract/introduction, as this approach in particular is of interest in the field, including the KNCV kit.

Response 1: Well accepted complement from the reviewer. In the revised version, note was made on the single step, centrifuge-free stool processing method adopted from the KNCV TB Foundation. 

Comment 2: There are some spacing issues, such as in line 61 “1.1million” and “in2018” in line 62. Please ensure spacing is okay throughout

Response 2: We accepted the reviewer’s comment. In the revised manuscript, this is corrected.

Comment 3: Were invalid stool results repeated? 

Response 3: No; due to shortage of Xpert cartridges, we were unable to repeat invalid stool-Xpert results. We have indicated this in the manuscript.

Comment 4: Please indicate in the results that none of the children with clinically-diagnosed TB (probable/possible) were stool-Xpert positive.

Response 4: We agree with reviewer’s comment and in the revised manuscript, it was indicated as suggested by the reviewer.

Comment 6: PLOS authors have the option to publish the peer review history of their article (what does this mean?). If published, this will include your full peer review and any attached files.

Response 6: We agree to publish the peer review process.

---

## [Decision Letter · Decision Letter 1]

7 Feb 2022

PONE-D-21-15359R1Stool-based Xpert MTB/RIF assay for the diagnosis of pulmonary tuberculosis in children at a Teaching and Referral Hospital in Southwest EthiopiaPLOS ONE

Dear Dr. Tadesse,

Thank you for submitting your manuscript to PLOS ONE. This manuscript is almost in publishable form. Therefore, we invite you to submit a revised version of the manuscript that addresses the points below.

We look forward to receiving your revised manuscript.

Kind regards,

Silvia S. Chiang

Academic Editor

PLOS ONE

Journal Requirements:

Additional Editor Comments:

PONE-D-21-15359-R1

"Stool-based Xpert MTB/RIF assay for the diagnosis of pulmonary tuberculosis in children at a Teaching and Referral Hospital in Southwest Ethiopia"

Apologies for the delayed review. I think this manuscript is almost in publishable form. This list of comments is very long because the journal does not have an English editing service, so I have edited the wording and phrasing myself. However, I did not edit the comma errors, and I suggest the authors ask a native English speaker to review and edit prior to resubmission.

In addition to the reviewer’s comment, I have the following additional minor edits:

1. To maintain consistency with the revised definition of the composite reference standard (CRS), which now includes “confirmed TB” and “unconfirmed TB” per the updated consensus case definitions by Graham, et al. and as recommended by the reviewer, please make the following changes:

a. Abstract, lines 30-32: Diagnostic performance was calculated with reference to LJ culture and to a composite reference standard (CRS) of confirmed TB (by Xpert or culture) and unconfirmed TB (clinical diagnosis with improvement after anti-TB treatment).

b. Study procedures, lines 115:117: Study subjects were diagnosed with confirmed TB based on positive microbiological test results (positive LJ culture and/or positive Xpert on gastric aspirate or sputum), or with unconfirmed TB based on a clinical diagnosis and positive response to ATT.

2. Data analysis, lines 173-174: Please modify this sentence based on the reviewer’s comment to use culture and/or Xpert positivity as one of the reference standards.

3. Results, lines 228-235 and relevant tables: Please modify based on the reviewer’s comment to use culture and/or Xpert positivity as one of the reference standards.

4. Background, line 60: Technically, SARS-CoV-2 has exceeded TB. I recommend saying “Tuberculosis (TB) is a global leading cause of death . . .

5. Discussion, lines 251-253: The lower sensitivity of stool Xpert compared to the CRS is because clinically diagnosed patients have paucibacillary disease. Would modify this sentence to be more precise. Also please use the abbreviation CRS to be consistent w/ the rest of the manuscript.

6. There are minor grammatical errors throughout the manuscript. I have asked PLOS One if there is an editing service, but unfortunately, there is not. I have listed some corrections to word choice below (underlined), but I suggest the authors ask a native English speaker to review the paper before resubmission. I did not correct errors in punctuation.

a. Abstract

i. Lines 21-22: . . . challenging mainly due to the difficulty of . . .

ii. Lines 29-30: . . . after application of a single-step, centrifuge-free stool processing method . . .

iii. Line 33: . . . 152 of whom had . . .

iv. Lines 35-36: . . . only ten had microbiologically confirmed (positive Xpert and/or culture) disease, and the remaining . . .

v. Lines 37-38: . . . compared to culture; however, the sensitivity . . .

vi. Lines 41-42: . . . sensitivity for stool is comparable to the sensitivity of Xpert for gastric aspirate . . .

b. Background

i. Lines 72-73: . . . TB endemic settings. Thus, there is a need for . . .

c. Study design and setting

i. Lines 94-96: . . . hospitals in the country. JUMC is a teaching and referral hospital that provides services for about 15 million people in its catchment area. Each year, JUMC reports approximately 15,000 inpatient hospital stays, 160,000 outpatient visits, and 11,000 emergency department visits.

ii. Lines 98-101: . . . an estimated annual incidence of 151,000 cases and an estimated 19,000 TB-related deaths, excluding HIV-associated TB deaths.

d. Specimen collection and laboratory procedures

i. Line 129: . . . for research purposes . . .

ii. Lines 132, 134 and all subsequent instances: Please write out “gastric aspirate” instead of using the abbreviation “GA” as this abbreviation is not commonly used and unnecessary (will be easier for readers to understand the paper without going back to look for what “GA” means)

iii. Line 138: . . . followed by vigorous shaking and incubation for 15 minutes . . .

iv. Line 145 and all subsequent instances: Please do not abbreviate “min”; please write out “minutes”

v. Line 153: . . . from KNCV TB Foundation.

e. Diagnostic classification for analysis

i. Lines 165-166: “CRS is defined as either confirmed TB or unconfirmed TB”; cases that met “unlikely TB” criteria were not classified as TB.

f. Data analysis

i. Lines 175-176: “Non-overlapping 95% CIs indicated a difference between the two methods (stool Xpert vs. gastric aspirate Xpert).”

g. Results

i. Line 189: . . . stool specimens.”

ii. Lines 189-191: Seventy-eight (51.3%) of 152 participants were females, and participants’ ages ranged from 7 months to 14 years (median 3 years). [Provide number here] (64.5%) of the participants were rural residents (Table 1).”

iii. Lines 192-197: Please report the number first, followed by the percentage in parentheses.

iv. Line 195: “The majority of children . . .”

v. Line 196: “Close to a third . . .”

h. Diagnosis of TB

i. Lines 200-206 and throughout the manuscript: Please be consistent and use either “M. tuberculosis” or “MTB”

ii. Line 208: I don’t believe PTB was defined previously. Please write out “pulmonary TB” if it was not.

iii. Lines 209-210: We reviewed the medical records of the remaining 142/152 (93.4%) of participants who were microbiologically negative.

i. Diagnostic performance of stool Xpert

i. Line 222: . . . a shortage of Xpert cartridges, we did not repeat . . .

j. Discussion

i. Line 239: The national guidelines for TB care . . .

ii. Line 241: respiratory specimens . . .

iii. Line 242: . . . for this population.

iv. Line 249: microbiologically confirmed and/or clinically diagnosed TB.

v. Line 283: . . . we did not assess the quality . . .

Reviewers' comments:

Reviewer's Responses to Questions

**Comments to the Author**

1. If the authors have adequately addressed your comments raised in a previous round of review and you feel that this manuscript is now acceptable for publication, you may indicate that here to bypass the “Comments to the Author” section, enter your conflict of interest statement in the “Confidential to Editor” section, and submit your "Accept" recommendation.

Reviewer #1: (No Response)

2. Is the manuscript technically sound, and do the data support the conclusions?

Reviewer #1: Yes

3. Has the statistical analysis been performed appropriately and rigorously? 

Reviewer #1: Yes

4. Have the authors made all data underlying the findings in their manuscript fully available?

Reviewer #1: Yes

5. Is the manuscript presented in an intelligible fashion and written in standard English?

Reviewer #1: Yes

6. Review Comments to the Author

Reviewer #1: I thank the authors for their thoughtful review of the comments, and have addressed the majority of them. For response 8, as the authors note culture alone will miss pediatric cases due to their paucibacillary disease. There are two goals of this analysis, one to provide an estimate of stool Xpert performance and the other to compare to respiratory specimen testing. In the first goal, the standard Confirmed TB definition should be used that includes Xpert results. For the second goal, it is reasonable to only use culture to allow comparison between the Xpert stool and GA to reduce bias. Would recommend presenting the results in these two ways; based on my understanding of the tables, this should still lead to 100% sensitivity and support their discussion line that all Xpert GA results were stool Xpert positive. Excluding Xpert from the main estimation results also confuses the comparison from microbiological to composite reference standard, as the CRS includes both culture and Xpert positive cases.

7. PLOS authors have the option to publish the peer review history of their article (what does this mean?). If published, this will include your full peer review and any attached files.

Reviewer #1: No

---

## [Author Response · Author response to Decision Letter 1]

6 Apr 2022

Response to Editor and Reviewers

We appreciate the editor and reviewers for the constructive comments which we have used to improve the quality of the manuscript. As usual, we have re-written some portions of the manuscript accordingly. We have also carefully addressed the comments line by line as follows.

Editor’s comments to the Author:

Response to Editor’s comments:

Journal Requirements:

Editor’s comment: Please review your reference list to ensure that it is complete and correct. If you have cited papers that have been retracted, please include the rationale for doing so in the manuscript text, or remove these references and replace them with relevant current references. Any changes to the reference list should be mentioned in the rebuttal letter that accompanies your revised manuscript. If you need to cite a retracted article, indicate the article’s retracted status in the References list and also include a citation and full reference for the retraction notice.

Response: We have checked the references list and confirmed that it is correct and complete. None of the reference cited was retracted. 

Additional Editor Comments 

Comment 1: English editing service of the whole manuscript.

Response 1: We are very much grateful for the editor for English editing services. We found it very helpful to improve the quality of our manuscripts. 

Additional minor edits from the Editor

Comment 1: To maintain consistency with the revised definition of the composite reference standard (CRS), which now includes “confirmed TB” and “unconfirmed TB” per the updated consensus case definitions by Graham, et al. and as recommended by the reviewer, please make the following changes: Comment 1a: Abstract, lines 30-32: Diagnostic performance was calculated with reference to LJ culture and to a composite reference standard (CRS) of confirmed TB (by Xpert or culture) and unconfirmed TB (clinical diagnosis with improvement after anti-TB treatment). 

Response 1a: We accepted the comment and corrected accordingly in the revised draft. Shown in Line 31-32 of cleaned version.

Comment 1b: Study procedures, lines 115:117: Study subjects were diagnosed with confirmed TB based on positive microbiological test results (positive LJ culture and/or positive Xpert on gastric aspirate or sputum), or with unconfirmed TB based on a clinical diagnosis and positive response to ATT.

Response 1b: We agree with the Editor’s suggestion and corrected accordingly. Shown in Line 116-118 of cleaned version.

Comment 2: Data analysis, lines 173-174: Please modify this sentence based on the reviewer’s comment to use culture and/or Xpert positivity as one of the reference standards.

Response 2: We found the editor’s comment important and corrected accordingly. This is shown in the data analysis (Line 175) and result sections (Table 4 and S1 Table). Line 235-237.

Comment 3: Results, lines 228-235 and relevant tables: Please modify based on the reviewer’s comment to use culture and/or Xpert positivity as one of the reference standards.

Response 3: The Editor’s/reviewer’s comment is clearly addressed in the revised version of the manuscript. This is shown in the data analysis (Line 175) and result sections (Table 4 and S1 Table). Line 235-237.

Comments 4: Background, line 60: Technically, SARS-CoV-2 has exceeded TB. I recommend saying “Tuberculosis (TB) is a global leading cause of death . . .

Response 4: It is modified as suggested by the Editor. Shown in Line 61 of cleaned version.

Comment 5: Discussion, lines 251-253: The lower sensitivity of stool Xpert compared to the CRS is because clinically diagnosed patients have paucibacillary disease. Would modify this sentence to be more precise. Also please use the abbreviation CRS to be consistent w/ the rest of the manuscript.

Response 5: We agree with the Editor’s comment and rewritten it accordingly. Look at Line 263-264 of cleaned-revised version.

Comment 6: There are minor grammatical errors throughout the manuscript. I have asked PLOS One if there is an editing service, but unfortunately, there is not. I have listed some corrections to word choice below (underlined), but I suggest the authors ask a native English speaker to review the paper before resubmission. I did not correct errors in punctuation. Comment 6a: Abstract

i. Lines 21-22: . . . challenging mainly due to the difficulty of . . .

ii. Lines 29-30: . . . after application of a single-step, centrifuge-free stool processing method . . .

iii. Line 33: . . . 152 of whom had . . .

iv. Lines 35-36: . . . only ten had microbiologically confirmed (positive Xpert and/or culture) disease, and the remaining . . .

v. Lines 37-38: . . . compared to culture; however, the sensitivity . . .

vi. Lines 41-42: . . . sensitivity for stool is comparable to the sensitivity of Xpert for gastric aspirate . . .

Comment 6b: Background

i. Lines 72-73: . . . TB endemic settings. Thus, there is a need for . . .

Comment 6c: Study design and setting

i. Lines 94-96: . . . hospitals in the country. JUMC is a teaching and referral hospital that provides services for about 15 million people in its catchment area. Each year, JUMC reports approximately 15,000 inpatient hospital stays, 160,000 outpatient visits, and 11,000 emergency department visits.

ii. Lines 98-101: . . . an estimated annual incidence of 151,000 cases and an estimated 19,000 TB-related deaths, excluding HIV-associated TB deaths.

Comment 6d: Specimen collection and laboratory procedures

i. Line 129: . . . for research purposes . . .

ii. Lines 132, 134 and all subsequent instances: Please write out “gastric aspirate” instead of using the abbreviation “GA” as this abbreviation is not commonly used and unnecessary (will be easier for readers to understand the paper without going back to look for what “GA” means)

iii. Line 138: . . . followed by vigorous shaking and incubation for 15 minutes . . .

iv. Line 145 and all subsequent instances: Please do not abbreviate “min”; please write out “minutes”

v. Line 153: . . . from KNCV TB Foundation.

Comment 6e: Diagnostic classification for analysis

i. Lines 165-166: “CRS is defined as either confirmed TB or unconfirmed TB”; cases that met “unlikely TB” criteria were not classified as TB.

Comment 6f: Data analysis

i. Lines 175-176: “Non-overlapping 95% CIs indicated a difference between the two methods (stool Xpert vs. gastric aspirate Xpert).”

Comment 6g: Results

i. Line 189: . . . stool specimens.”

ii. Lines 189-191: Seventy-eight (51.3%) of 152 participants were females, and participants’ ages ranged from 7 months to

iii. 14 years (median 3 years). [Provide number here] (64.5%) of the participants were rural residents (Table 1).”

iv. Lines 192-197: Please report the number first, followed by the percentage in parentheses.

v. Line 195: “The majority of children . . .”

vi. Line 196: “Close to a third . . .”

Comment 6h: Diagnosis of TB

i. Lines 200-206 and throughout the manuscript: Please be consistent and use either “M. tuberculosis” or “MTB”

ii. Line 208: I don’t believe PTB was defined previously. Please write out “pulmonary TB” if it was not.

iii. Lines 209-210: We reviewed the medical records of the remaining 142/152 (93.4%) of participants who were microbiologically negative.

Comment 6i: Diagnostic performance of stool Xpert

i. Line 222: . . . a shortage of Xpert cartridges, we did not repeat . . .

Comment 6j. Discussion

ii. Line 239: The national guidelines for TB care . . .

iii. Line 241: respiratory specimens . . .

iv. Line 242: . . . for this population.

v. Line 249: microbiologically confirmed and/or clinically diagnosed TB.

vi. Line 283: . . . we did not assess the quality . . .

Response 6 (6a-6j): We are very grateful to the editor for his/her time and for correcting grammatical errors throughout the manuscript. The feedback from the editor is well accepted and considered in the revised version of the manuscript. We also shared our last version of the manuscript with language expertise and they made substantial English Language editions. 

Review #1 Comments to the Author

Response to Reviewers’ comments:

Comment 1: I thank the authors for their thoughtful review of the comments, and have addressed the majority of them. For response 8, as the authors note culture alone will miss pediatric cases due to their paucibacillary disease. There are two goals of this analysis, one to provide an estimate of stool Xpert performance and the other to compare to respiratory specimen testing. In the first goal, the standard Confirmed TB definition should be used that includes Xpert results. For the second goal, it is reasonable to only use culture to allow comparison between the Xpert stool and GA to reduce bias. Would recommend presenting the results in these two ways; based on my understanding of the tables, this should still lead to 100% sensitivity and support their discussion line that all Xpert GA results were stool Xpert positive. Excluding Xpert from the main estimation results also confuses the comparison from microbiological to composite reference standard, as the CRS includes both culture and Xpert positive cases.

Response 1: We found that the feedback from the reviewer #1 is very interesting and the comment is addressed in method and results sections. In S1 Table (supporting information), stool Xpert and GA Xpert MTB detection rate compared to LJ culture and/or GA Xpert positivity (microbiological confirmation) was described. Moreover, the diagnostic performances of stool Xpert and GA Xpert using LJ culture and/or GA Xpert (microbiological confirmations) as one of the reference standards was indicated in Table 4. This is shown in the data analysis (Line 175) and result sections (Table 4 and S1 Table). Line 235-237. 

Comment 2: 7. PLOS authors have the option to publish the peer review history of their article (what does this mean?). If published, this will include your full peer review and any attached files.

Response 2: We agree to publish the peer review process. 

Comment 3: While revising your submission, please upload your figure files to the Preflight Analysis and Conversion Engine (PACE) digital diagnostic tool, https://pacev2.apexcovantage.com/. PACE helps ensure that figures meet PLOS requirements. To use PACE, you must first register as a user. Registration is free. Then, login and navigate to the UPLOAD tab, where you will find detailed instructions on how to use the tool. If you encounter any issues or have any questions when using PACE, please email PLOS at figures@plos.org. Please note that Supporting Information files do not need this step. 

Response 3: We have checked our figure by PACE and found that it met PLOSONE requirements.

---

## [Editor Report · Decision Letter 2]

13 Apr 2022

Stool-based Xpert MTB/RIF assay for the diagnosis of pulmonary tuberculosis in children at a teaching and referral hospital in Southwest Ethiopia

PONE-D-21-15359R2

Dear Dr. Tadesse,

We’re pleased to inform you that your manuscript has been judged scientifically suitable for publication and will be formally accepted for publication once it meets all outstanding technical requirements.

Kind regards,

Silvia S. Chiang

Academic Editor

PLOS ONE
---

## [Editor Report · Acceptance letter]

26 Apr 2022

PONE-D-21-15359R2 

Stool-based Xpert MTB/RIF assay for the diagnosis of pulmonary tuberculosis in children at a teaching and referral hospital in Southwest Ethiopia 

Dear Dr. Tadesse:

I'm pleased to inform you that your manuscript has been deemed suitable for publication in PLOS ONE. Congratulations! Your manuscript is now with our production department. 

Kind regards, 

on behalf of

Dr. Silvia S. Chiang 

Academic Editor

PLOS ONE